# Effects of an EPS Biosynthesis Gene Cluster of *Paenibacillus polymyxa* WLY78 on Biofilm Formation and Nitrogen Fixation under Aerobic Conditions

**DOI:** 10.3390/microorganisms9020289

**Published:** 2021-01-30

**Authors:** Xiaojuan He, Qin Li, Nan Wang, Sanfeng Chen

**Affiliations:** 1State Key Laboratory of Agrobiotechnology and College of Biological Sciences, China Agricultural University, Beijing 100193, China; hexiaojuan2013@163.com (X.H.); lqliqin1@126.com (Q.L.); 2Biotechnology Research Institute, Chinese Academy of Agricultural Sciences, Beijing 100081, China; wangnan@caas.cn

**Keywords:** exopolysaccharides, biofilm, nitrogen fixation, *Paenibacillus polymyxa*

## Abstract

Exopolysaccharides (EPS) are of high significance in bacterial biofilm formation. However, the effects of EPS cluster(s) on biofilm formation in *Paenibacillus* species are little known. In this study, we have shown that *Paenibacillus polymyxa* WLY78, a N_2_-fixing bacterium, can form biofilm. EPS is the major component of the extracellular matrix. The genome of *P. polymyxa* WLY78 contains two putative gene clusters (designated *pep-1* cluster and *pep-2* cluster). The *pep-1* cluster is composed of 12 putative genes (*pepO-lytR*) co-located in a 13 kb region. The *pep-2* cluster contains 17 putative genes (*pepA-pepN*) organized as an operon in a 20 kb region. Mutation analysis reveals that the *pep*-2 cluster is involved in EPS biosynthesis and biofilm formation. Disruption of the *pep*-2 cluster also leads to the enhancement of motility and change of the colony morphology. In contrast, disruption of the *pep-1* cluster does not affect EPS synthesis or biofilm formation. More importantly, the biofilm allowed *P. polymyxa* WLY78 to fix nitrogen in aerobic conditions, suggesting that biofilm may provide a microaerobic environment for nitrogenase synthesis and activity.

## 1. Introduction

Bacteria can obtain many survival benefits by forming biofilm. Biofilms are architecturally complex communities of microorganisms in which the cells are held together by an extracellular matrix [1,2]. The extracellular matrix is typically composed of exopolysaccharides (EPSs), proteins, and extracellular DNA (eDNA) [3]. Many studies have focused on the biofilm formation of pathogens, such as *Pseudomonas aeruginosa*, *Streptococcus mutans*, *Escherichia coli*, etc [4,5,6]. Recent reports have highlighted the biofilm formation of the beneficial bacteria in agriculture, such as the plant growth-promoting bacteria *Bacillus subtilis*, *Azospirillum brasilense*, *Rhizobia*, etc [7,8,9,10]. 

*B. subtilis* is a model organism for studying the mechanisms of biofilm [11]. Many studies showed that EPS play a critical role in biofilm formation [12]. A 15-gene *eps* operon *epsA-O*) and a *tapA-sipW-tasA* operon in *B. subtilis* are required for extracellular polysaccharide synthesis and *sinR* regulates *epsA-O* and *tapA-sipW-tasA* operons [13,14,15]. An *eps* cluster composed of 18 putative open reading frames (ORFs) encodes enzymes for the production of a high molecular-weight heteropolysaccharide, exopolysaccharides (EPS) in *Lactobacillus paraplantarum* BGCG11 [16]. The extracellular polysaccharides of rhizobacteria (e.g. *Sinorhizobium meliloti* and *Azospirillum brasilense* Sp7) play an important role in biofilm formation and the interactions between bacterium and plant roots [17,18,19]. The surface polysaccharides of *Rhizobia* are critical for attachment to plant roots and biofilm formation [18]. 

Some species of the *Paenibacillus* genus are the important plant growth-promoting rhizobacteria (PGPR). The biofilms produced by some *Paenibacillus* species can effectively help them to colonize on plant roots and to help host plants to adapt and survive in harsh environments [20,21]. For example, two naturally separated *P. polymyxa* B1 and B2 form biofilm on root tips, protecting the plant against pathogens and abiotic stress in *Arabidopsis thaliana* [20]. The biofilm polysaccharides of *P. polymyxa* A26 are capable of antagonizing *Fusarium graminearum* [21]. In addition, it was reported that *Paenibacillus lentimorbus* biofilms alleviate drought stress in chickpea [22]. Because extracellular polysaccharides are the key component of biofilm, their production also contributes to the successful colonization of bacteria in plant roots [23]. Moreover, *Paenibacillus* spp. were discovered to produce various EPS with different biotechnological properties that can be used in antitumor [24], antioxidant [24,25] or flocculating activity [26,27]. Although there is a growing interest in EPS biosynthesis and biofilm formation of *Paenibacillus* spp., little is known about the mechanisms of biofilm formation. One reason for the slow progress in the development of mechanisms may be that most *Paenibacillus* spp. struggle to break through genetic transformation [28]. 

Nitrogen fixation is one of the strategies used by PGPR to promote plant growth. Biological nitrogen fixation, the conversion of atmospheric N_2_ to NH_3_, is mainly catalyzed by molybdenum-dependent nitrogenase, which is distributed within bacteria and archaea [29]. Nitrogenase consists of two metalloprotein components, the Fe protein (encoded by *nifH*) and the MoFe protein (encoded by *nifD* and *nifK*), which contain different metal clusters. Nitrogenase is oxygen sensitive and requires ATP for activity [30]. Synthesis and activity of nitrogenase are tightly regulated in response to the levels of fixed nitrogen, carbon, energy and the external oxygen concentration [30]. *P. polymyxa* WLY78 is a N_2_-fixing bacterium isolated by our laboratory and this bacterium performs nitrogen fixation under anaerobic or microaerobic conditions [31]. This bacterium also could promote plant growth [32]. However, whether *P. polymyxa* WLY78 can form biofilm and whether biofilm is beneficial for nitrogen fixation are not known. 

In this study, we find that the genome of *P. polymyxa* WLY78 contains two putative gene clusters (designated *pep-1* cluster and *pep-2* cluster). The *pep-1* cluster is composed of 12 putative genes (*pepO-ltyR*) and the *pep-2* cluster contains 17 putative genes (*pepA-pepN*). Two mutants ∆*pep-1* and ∆*pep-2* were constructed by deletion of the 11 genes (*pepO-ugdH2*) located within the *pep-1* gene cluster and by deletion of the 15 genes (*pepA-manC*) located within the *pep-2* gene cluster, respectively. Then, wild-type *P. polymyxa* WLY78, ∆*pep-1* and ∆*pep-2* mutants were comparatively investigated in terms of their abilities in EPS production, biofilm formation, morphology, motility. Our results revealed that the *pep*-2 cluster is responsible for EPS biosynthesis and biofilm formation and the biofilm provides a microaerobic environment for nitrogen fixation. These findings will provide guidance for improving the efficiency of colonization and nitrogen fixation in natural conditions of *P. polymyxa* WLY78 by enhancing production of EPS and biofilm.

## 2. Materials and Methods 

### 2.1. Strains, Plasmids, Primers, and Media

Bacterial strains, plasmids and primers used in this study are listed in Appendix A. Luria–Bertani (LB) was used for enrichment culture. For nitrogenase activity assay in biofilms, *P. polymyxa* WLY78 and its mutants were grown in nitrogen-free medium (per liter: 10.4 g Na_2_HPO_4_, 3.4 g KH_2_PO_4_, 26 mg CaCl_2_⋅2H_2_O, 30 mg MgSO_4_, 0.3 mg MnSO_4_, 36 mg Ferric citrate, 7.6 mg Na_2_MoO_4_⋅2H_2_O, 10 μg p -aminobenzoic acid, 5 μg biotin) supplemented with 200 mM glucose and 10 mM glutamate [33]. For biofilm formation assay, nitrogen-free medium supplemented with 200 mM glucose and 20 mM NH_4_Cl was used. *P. polymyxa* WLY78 and its mutants were grown at 30 °C. *Escherichia coli* strain JM109 was used as routine cloning and was grown in LB broth at 37 °C with shaking at 200 rpm. Thermo-sensitive vector pRN5101, which has ampicillin and erythromycin resistance, was used for gene disruption in *P. polymyxa* [34]. When required, 100 μg/mL ampicillin and 5 μg/mL erythromycin were used. 

### 2.2. Construction of ∆pep-1 and ∆pep-2 Mutants

The two mutants were constructed by a homologous recombination method. The primers used for PCR were listed in Appendix A. For disruption of the *pep-1* cluster, about 1 kb upstream fragment flanking *ugdH2* was obtained by PCR with primers pep-1-up-F and pep-1-up-R, and 1 kb downstream fragment flanking *pepO* was obtained by PCR with primers pep-1-down-F and pep-1-down-R from the genomic DNA of *P. polymyxa* WLY78. The two fragments were then assembled with the *BamH*Ⅰ/*Hind*Ⅲ digested pRN5101 vector by Gibson assembly master mix (New England, USA Biolabs) and the recombinant plasmid pRDpep-1 was screened by ampicillin resistance. Similarly, for disruption of the *pep-2* cluster, about 1 kb upstream fragment flanking *manC* was obtained by PCR with primers pep-2-up-F and pep-2-up-R and 1 kb downstream fragment flanking *pepA* was obtained by PCR with primers pep-2-down-F and pep-2-down-R from the genomic DNA of *P. polymyxa* WLY78. The two fragments were then assembled with the *BamH*Ⅰ/*Hind*Ⅲ digested pRN5101 vector and the recombinant plasmid pRDpep-2 was screened by ampicillin resistance. Then, the two recombinant plasmids were transformed into *P. polymyxa* WLY78, respectively, and the single crossover transformants were chosen through erythromycin resistance. Subsequently, deletion mutants (the double-crossover transformants) ∆*pep-1* and ∆*pep-2* were selected after several rounds of nonselective growth at 39 °C and confirmed by PCR amplification and analyzing sequence. The ∆*pep-1* mutant wan confirmed by PCR amplifying a 2072 bp DNA fragment with primers pep-1-up-F and pep-1-down-R, while the ∆*pep-2* mutant was confirmed by PCR amplifying a 2102 bp DNA fragment with primers pep-2-up-F and pep-2-down-R. Then, these PCR fragments were sequenced.

### 2.3. Biofilm Assay

For the biofilm quantification assay, *P. polymyxa* WLY78 and its mutants were cultivated in a glass tube containing 1 mL of nitrogen-free medium supplemented with 200 mM glucose and 20 mM NH_4_Cl at 30 °C without agitation for 72–96 h. To determine the effects of the different nitrogen sources on biofilm formation, 0–100 mM NH_4_Cl or 0–40 mM glutamate was used with 200 mM glucose in the liquid–air interface biofilm assay at 30 °C without agitation for 96 h. To compare the effects of the different carbon sources on biofilm formation, 200 mM carbon sources, such as glucose, sucrose, lactose, starch, sodium, potassium acetate and glycerinum were used with 20 mM NH_4_Cl as nitrogen sources in the liquid–air interface biofilm assay at 3 0°C without agitation for 96 h. Then, 0–500 mM glucose was added with 20 mM NH_4_Cl to test the effects on biofilm formation. To observe biofilm formation, the cells that were not involved in biofilm formation were sucked out and washed twice, a final concentration of 0.1% Crystal Violet was added to each test tube, and the tube was incubated at room temperature for 10 min before rinsing in distilled water. To quantify biofilm formation, the biofilm stained with Crystal Violet was dissolved 1 mL 40% acetic acid and was determined by measuring the absorption value at OD_570_.

For the air–solid biofilm (colony biofilm) assay, the same medium as the liquid–air interface biofilm assay was used for preparing agar plates, with the addition of 1.5% agar. The overnight culture solution was adjusted to OD_600_ =1. Then, 5 μL of culture was drawn to the plates. The plates were cultivated for 120 h at 30 °C for observation. All quantification assays were performed at least in triplicate. 

### 2.4. Isolation, Extraction and Analysis of EPS

The cultures were harvested and centrifuged at 8000 rpm for 15 min and the supernatants were processed for EPS extraction. The sediment was then dried until a constant weight was achieved and then weighed. The EPS from the supernatant was extracted by adding twofold volumes of ice-cold ethyl alcohol with mixing followed by overnight storage at 4 °C. Precipitated EPS was collected by centrifugation at 8000 rpm, 4 °C for 30 min. After drying, the extracted EPS was dissolved in deionized water [35,36]. 

Carbohydrate was measured according to the Phenol–Sulfuric acid method [37] and glucose was served as the standard. Soluble protein was measured by the method of Lowry [37] and bovine serum albumin was served as the standard. The eDNA was extracted once by adding an equal volume of phenol/chloroform/isoamyl alcohol (25:24:1). Then, the DNA of the aqueous phase was precipitated by 3 vol of ice-cold 100% (*v/v*) ethanol and dissolved in deionized water [38].

### 2.5. Colonial Morphology and Motility Assays

For observation of colony morphology, *P. polymyxa* WLY78 or its mutant was cultivated on LB solid plate (LB plus 1.5% agar) overnight at 37 °C. Flagellum-mediated motility assays were implemented by spotting a single colony on 0.7% LB solid medium. After 24 h of incubation at 37 °C, the motility was evaluated by examining the distance of colonies spread beyond the inoculation site [39].

### 2.6. Acetylene Reduction Assay for Nitrogenase Activity

For measuring the nitrogenase activity of bacteria under planktonic and anaerobic conditions, nitrogen-deficient medium containing 2 mM glutamate as the nitrogen sources and 20 mM glucose as carbon sources were used. *P. polymyxa* WLY78 or its mutant was grown in 5 mL of LB and then inoculated in 50 mL flasks with shaking at 200 rpm overnight at 30 °C. The culture was collected by centrifugation, washed three times with sterilized water and then resuspended in nitrogen-deficient medium supplemented with glutamate and glucose. The final OD_600_ was 0.4–0.5. Then, 4 mL of the culture was transferred to 26-mL test tubes sealed with a rubber stopper. The headspace in the tubes was then evacuated and replaced with argon gas and, at the same time, 10% (*v/v*) acetylene was added. After incubating the cultures for 20–24 h at 30 °C with shaking at 200 rpm, ethylene production was analyzed by gas chromatography. Nitrogenase activity was expressed in nmol C_2_H_4_/mg protein/h [33].

To examine the nitrogenase activity of bacteria in the biofilm formed with different oxygen concentrations, *P. polymyxa* WLY78 or its mutant was grown in glass tube filled with liquid medium and simultaneously different oxygen concentrations was injected (nitrogen-free medium supplemented with 20 mM NH_4_Cl and 200 mM glucose) without agitation. After about 96 h of cultivation, the biofilm should be formed. Then, 10% (*v/v*) acetylene were injected into the test tubes through rubber stopper. After 12–14 h of incubation at 30 °C without shaking, ethylene was measured [40].

### 2.7. Statistical Analysis

Statistical tests were performed using SPSS software version 20 (SPSS Inc., Chicago, IL, United States). Two-way analysis of variance (ANOVA) was employed to check the significant differences between treatments. Means of different treatments were compared using the least significant difference (LSD) at the 0.05, 0.01 or 0.001 level of probability. Graphs were prepared using GraphPad Prism software v. 8.0 (GraphPad Software Inc., San Diego, CA, USA).

## 3. Results

### 3.1. The Effects of Carbon Sources and Nitrogen Sources on Biofilm in P. polymyxa WLY78

Biofilm formation of *P. polymyxa* WLY78 was investigated by growing the bacterial cells in the static liquid culture at 30 °C for 0–120 h and by using the liquid–air interface biofilm assay. We found that the highest amount of biofilm was produced after 96 h of cultivation in nitrogen-free medium supplemented with 200 mM glucose and 20 mM NH_4_Cl (Figure 1A). In the absence of nitrogen, a small amount of biofilm was produced, while a high amount of biofilm was formed at 20–40 mM NH_4_Cl or at more than 10 mM glutamate (Figure 1B,C). The highest amount of biofilm was produced when glucose was used as a carbon source, followed by sucrose, starch, glycerinum and lactose in order, whereas nearly no biofilm was formed in the presence of sodium citrate or potassium acetate (Figure 1D). The highest amount of biofilm was produced when 200 mM glucose was used (Figure 1E). These results suggested that sufficient carbon sources and nitrogen sources were critical for the *P. polymyxa* WLY78 biofilm formation.

In order to detect the component contents of the extracellular matrix in *P. polymyxa* WLY78, the extracellular complex was isolated from the shaking bacterial culture. The content of polysaccharides in the extracellular matrix was about 90%, whereas the protein content was about 2% and the eDNA content was around 7% (Figure 1F). These results indicate that EPS was the major component of the extracellular matrix.

### 3.2. Identification of Polysaccharide Biosynthesis Gene Clusters in P. polymyxa WLY78

Here, we searched the genome of *P. polymyxa* WLY78 sequenced by our laboratory [32] and found that there was a large gene cluster coding for EPS production which shared very high similarity with that of *P. polymyxa* DSM365. This EPS cluster comprises 29 putative genes on a ~36.8 kb DNA fragment [41]. The 13 kb region carrying 12 putative genes (*pepO-lytR*) is here designated as the *pep-1* gene cluster, and the 20 kb region containing 17 putative genes (*pepA*-*pepN*) is named the *pep-2* gene cluster. The *pep-1* gene cluster and the *pep-2* gene cluster are in the same transcription direction, and there is a ~2 kb space region containing a gene in the reverse orientation (arrow marked blue) between the two gene clusters (Figure 2A). Among the 12 putative genes within the *pep-1* gene cluster, with the exception of there being a longer interval region (1.98 kb) between *pepS* and *pepT*, other genes are overlapped or have space regions shorter than 86 bp. Among the 17 putative genes within the *pep-2* gene cluster, the longest interval region is 574 bp between *manC* and *ugdH1*, followed by 217 bp between *pepM* and *manC* and 121 bp between *pepM* and *pepN,* and space regions that overlap or are shorter than 65 bp are found among other ORFs. Furthermore, RT-PCR experiments using primers designed to span across intergenic regions revealed that *pepM*, *pepN*, *manC*, *ugdH1* and other genes were co-transcribed (Figure 2B), indicating that the 17 putative genes within the *pep-2* gene cluster are organized as an operon, whereas RT-PCR analysis showed no co-transcription between *pepN* and *pepO,* supporting that they belong to two different gene clusters (Figure 2B). 

The predicted functions of the 29 putative genes within the *pep-1* and *pep-2* gene clusters are listed in Table 1. The *ugdH1* and *manC* are probably involved in the synthesis of nucleoside sugar precursors [41]. The eight genes *galU*, *pepC*, *pepD*, *pepF*, *pepI*, *pepJ*, *pepK* and *pepL* are putative glycosyltransferases (GTs), which may transfer defined sugar moieties to the nascent, pre-assembled repeating units. The *pepH* or *pepR* is predicted to encode flippase and is likely involved in the translocation of sugar moieties from the cytoplasm to the outside of membrane [42]. The predicted function of pepE and pepG is the role of polymerase in the polymerization of sugars [43]. Furthermore, PepA and PepB, PepO and PepP were likely to regulate the length of polysaccharide chains [44]. PepM and PepN, whose functions are predicted to be glycosyl hydrolases, have the potential to hydrolyze its polysaccharide. This provides the basis for our following research of the *pep-2* gene cluster.

### 3.3. The pep-2 Gene Cluster Is Involved in Biofilm Formation and Polysaccharide Biosynthesis

To investigate the roles of the EPS gene clusters of *P. polymyxa* WLY78 in biofilm formation and polysaccharide biosynthesis, the *pep-1* and *pep-2* gene clusters were, respectively, knocked-out by using the homologous recombination method. The ∆*pep-1* mutant was constructed by removing the 11 genes (*pepO-ugdH2*) with a length of 12.6 kb DNA, while the ∆*pep-2* mutant was constructed by removing the 15 genes (*pepA-manC*) with a length of 16.5 kb DNA. The two mutants were confirmed by PCR with primers used for homologous recombination described in materials and methods, and an amplicon of about 2 kb was observed in the ∆*pep-1* mutant and the ∆*pep-2* mutant, respectively (Appendix A), and then the two amplicons were sequenced. 

The growth rate, biofilm formation ability and EPS content of *P. polymyxa* WLY78, ∆*pep-1* and ∆*pep-2* were comparatively analyzed when they were cultivated in liquid medium. As shown in Figure 3A, there was no obvious difference in growth rates among ∆*pep-1*, ∆*pep-2* and *P. polymyxa* WLY78. Compared to wild-type *P. polymyxa* WLY78, the ∆*pep-2* mutant produced very low amounts of biofilm and EPS and the ∆*pep-1* mutant did not show a significant difference in biofilm biomass and EPS content (Figure 3B,C). In addition, the major component of the extracellular matrix of the ∆*pep-2* mutant was eDNA with low amounts of EPS and protein (Figure 3D). The data suggest that the *pep-2* gene cluster is responsible for polysaccharide biosynthesis, which is essential for biofilm formation, while the *pep-1* gene cluster is not. Further, EPS was purified from *P. polymyxa* WLY78 and then different amounts of EPS were added to the cultures of the ∆*pep-2* mutant. The addition of EPS purified from *P. polymyxa* WLY78 could not help the ∆*pep-2* mutant to form biofilm (Appendix A). 

### 3.4. EPS Biosynthesis Defect Is Effect on Colony Biofilm, Colony Morphology and Motility

The colony biofilm, morphology and twitching motility of *P. polymyxa* WLY78, ∆*pep-1* and ∆*pep-2* were comparatively investigated. The air–solid biofilm assay showed that ∆*pep-1* mutant produced a thick colony biofilm as *P. polymyxa* WLY78 did, while ∆*pep-2* mutant nearly did not form colony biofilm (Figure 4A)**.** Both *P. polymyxa* WLY78 and ∆*pep-1* mutant formed a thick, sticky and white colony, whereas the *pep-2* mutant formed a non-adhesive, thin and partially transparent colony on Luria Bertani (LB) agar (Figure 4B). Motility of the ∆*pep-2* was significantly higher than those of *P. polymyxa* WLY78 and ∆*pep-1* mutant (Figure 4C). The data suggest the colony biofilm formation and mobility were influenced by the defects of EPS. This conclusion is consistent with the report that the bacterial motility affecting biofilm formation and colony morphology is also related to the secretion of extracellular matrix [45].

### 3.5. Biofilm Formation Mediated by EPS Enables Bacteria to Fix Nitrogen in Air

Nitrogenase is very sensitive to oxygen. Like almost all the N_2_-fixing microorganisms, *P. polymyxa* WLY78 fixes nitrogen only under anaerobic or microaerobic conditions [31]. As shown in Figure 5A, the nitrogenase activity of *P. polymyxa* WLY78 in biofilms was significantly higher than that of the ∆*pep-2* mutant under aerobic conditions, whereas the nitrogenase activity of the planktonic cells of *P. polymyxa* WLY78 and the ∆*pep-2* mutant in shaking cultures did not display distinct difference under anaerobic conditions. The results suggested that biofilm formation mediated by EPS enabled *P. polymyxa* WLY78 to fix nitrogen under aerobic conditions. We deduced that biofilm formation may provide a suitable microaerobic environment for nitrogenase synthesis and activity, which was consistent with the findings in *Pseudomonas stutzeri* A1501 [40]. 

The expression of nitrogen-fixing genes in the biofilm state under aerobic conditions and in the planktonic state under anaerobic conditions was further examined. Our recent studies have revealed that a minimal and compact *nif* gene cluster comprising nine genes (*nifB nifH nifD nifK nifE nifN nifX hesA nifV*) encoding Mo-nitrogenase is observed in *P. polymyxa* WLY78 [33]. To examine the effect of *pep-2* mutation on the transcription of *nif* genes in *P*. *polymyxa* WLY78, the transcript levels of *nifH*, *nifD* and *nifK* were determined by qRT-PCR. Under anaerobic condition in the planktonic state, the ∆*pep-2* mutant showed high transcript levels of the *nifHDK* compared to *P*. *polymyxa* WLY78. Under aerobic condition in the biofilm state, *P*. *polymyxa* WLY78 exhibited high transcript levels of the *nifHDK*, but the ∆*pep-2* mutant only showed the basal transcript levels (Figure 5C). The data are consistent with the above described nitrogenase activities. These results suggest that biofilm provides micro-aerobic conditions for nitrogen fixation.

### 3.6. Impacts of Nitrogen and Oxygen Content on Nitrogenase Activity in Biofilm

Here, we tested the effect of oxygen concentration on biofilm formation and nitrogenase activity in biofilms. Assays for biofilm formation and nitrogenase activity of *P. polymyxa* WLY78 were simultaneously performed after growing the bacterial cells in the static liquid culture at 30 °C for 7 days. Only a small amount of biofilm was formed in the absence of oxygen, and the highest amount of biofilm was produced when oxygen concentration was 21% (aerobic condition) (Figure 6A). In contrast, the highest nitrogenase activity was observed in absence of oxygen, in agreement with our previous reports that this bacterium exhibited the highest nitrogenase activity in anaerobic conditions [33]. Notably, nitrogenase activity was observed in the condition of 21% oxygen where biofilm was produced (Figure 6B), suggesting that biofilm provides the micro-aerobic conditions for nitrogenase. 

Our previous study showed that *P. polymyxa* WLY78 had the highest nitrogenase activity in the absence of NH_4_Cl and had no activity in the presence of more than 5 mM NH_4_Cl [46]. In this study, the effects of biofilm formation on the nitrogenase activity were investigated by growing *P. polymyxa* WLY78 in the presence of 21% oxygen and in nitrogen-free liquid medium supplemented with 200 mM glucose and different concentrations of nitrogen (0, 5, 10, and 20 mM glutamate or NH_4_Cl) at 30 °C for 7 days without shaking. As shown in Figure 6C, the highest nitrogenase activity was observed at 5 mM NH_4_Cl or at 10 mM glutamate, but the activity was reduced as the concentration of NH_4_Cl or glutamate increased. The current data suggest that biofilm formation could slightly enhance the resistance of nitrogenase to ammonium inhibition, but biofilm could not eliminate the inhibition of high concentrations of nitrogen on nitrogenase activity. 

Taken together, *P. polymyxa* WLY78 can form biofilm at the air–liquid interface and the colony biofilms on solid medium. EPS are the major component of the extracellular matrix. The *pep-2* operon, comprising 17 putative genes (*pepA*-*pepN*), is involved in EPS biosynthesis and biofilm formation, but the *pep-1* cluster composed of 12 genes (*pepO*-*ltyR*) is not. Biofilm formation mediated by EPS enables *P. polymyxa* WLY78 to fix nitrogen under aerobic conditions (Figure 7). 

## 4. Discussion 

In this study, we demonstrated that *P. polymyxa* WLY78 formed biofilm in standing liquid culture and colony biofilm on solid medium and EPS was the major component of the extracellular matrix. We searched the genome of *P. polymyxa* WLY78 and found two putative gene clusters responsible for EPS biosynthesis (designated *pep-1* cluster and *pep-2* cluster). The *pep-1* cluster is composed of 12 putative genes (*pepO-lytR*) and the *pep-2* cluster contains 17 putative genes (*pepA-pepN*). The two *pep* clusters were separated by ~2 kb. The combination of gene organization and RT-PCR showed that the 17 genes within the *pep-2* cluster were organized as an operon. Many bacteria that form biofilm have been reported to own at least one large polysaccharide gene cluster. For example, the EPS gene cluster of *Lactobacillus rhamnosus* comprises 17 ORFs co-located in chromosomal DNA regions of 14 to 18 kb [47]. A 15-gene *eps* operon (*epsA-O*) in *B. subtilis* was responsible for EPS synthesis [13,14,15]. Mutation analysis showed that only the *pep-2* cluster was required for EPS biosynthesis and biofilm formation in *P. polymyxa* WLY78. The report that the main components (glucose, mannose and galactose) of exopolysaccharide in *P. polymyxa* DSM 365 were no longer synthesized when the EPS synthesis gene cluster was mutated also confirms our results [41]. 

The functions of each gene within the 15-gene *eps* operon (*epsA-O*) in *B. subtilis* was well studied. In contrast, the functions of the putative genes in the *pep* clusters of *P. polymyxa* WLY78 in biofilm formation are unknown. *B. subtilis* EpsA, the homologous protein of PepA, was reported to phosphorylate and regulate many proteins involved in the synthesis of fatty acid and the metabolism of DNA and carbon [48]. *B. subtilis* EpsB, the homologous protein of PepB (tyrosine protein kinase), controls the process of exopolysaccharide production [48]. In *B. subtilis*, the activity of EpsB is stimulated by its modulator EpsA [49]. As the homologous protein of PepD and GalU, EpsE (glycosyltransferase) has the dual functions of inhibiting motility and participation in EPS production, and their synergism promotes the formation of *B. subtilis* biofilm [50]. Expression of the *epsA-O* operon in *B. subtilis* was negatively regulated by SinR [7,51]. Spo0A governed the regulatory pathway for matrix gene expression by influencing the activity of the master regulator SinR [52]. All these provide some ideas for further study of EPS regulation and the function of each gene.

Compared to wild-type *P. polymyxa* WLY78, the ∆*pep-2* mutant had enhanced motility. Explanation of this phenomenon was that a mass of EPS made bacteria stick together and thus inhibited bacterium motility. In addition, we do not know whether *P. polymyxa* PepD and GalU inhibit motility as their homologous protein of *B. subtilis* EpsE did. Previous studies showed that flagellum-mediated motility is required for biofilm growth through recruiting the cells of planktonic phase in *Listeria monocytogenes* [53]. It was also found that reaching the surfaces of biofilm was dependent on motility in *P. aeruginosa*, *E. coli* and *Bacillus cereus* [54,55]. However, there was evidence that the influence of motility relied on culture conditions and the production of exopolysaccharides in *P. aeruginosa* [56] and *Agrobacterium tumefaciens* [57].

Since nitrogenase is very sensitive to oxygen, nearly all N_2_-fixing microorganisms, including *P. polymyxa* WLY78, perform nitrogen fixation under anaerobic or microaerobic conditions. This study revealed that the cells of *P. polymyxa* WLY78 in biofilm had nitrogenase activity in air, but the ∆*pep-2* mutant did not. The data suggest that biofilm formation mediated by EPS could overcome the inhibition of oxygen on nitrogenase activity. The explanation of this phenomenon was that a mass of EPS could encase bacterial aggregate and it acted as a barrier to provide an anaerobic environment for nitrogenase. The similar phenomenon that the biofilm could enable bacteria to fix nitrogen in air was observed in *P. stutzeri*, *A. brasilense* and *Azotobacter vinelandii* [40,58]. Thus, it may be worth studying how to increase EPS synthesis and biofilm production without inhibiting nitrogenase activity in air. 

*Paenibacillus* and *Bacillus* are two genera of firmicutes. However, homology analysis showed that the 17-gene *pep-2* cluster of *P. polymyxa* WLY78 had very low similarity with the 15-gene *eps* operon of *B. subtilis*, suggesting that the two EPS synthesis clusters may originate differently. Synthesis and regulation mechanisms of biofilm formation in *B. subtilis* have been well studied. Although there is a growing interest in EPS biosynthesis and biofilm formation of *Paenibacillus,* the studies on regulation mechanisms of biofilm in *Paenibacillus* strains are very few. Our study will provide the foundation for studying the regulation of EPS biosynthesis and biofilm formation in *Paenibacillus*.

## Figures and Tables

**Figure 1 microorganisms-09-00289-f001:**
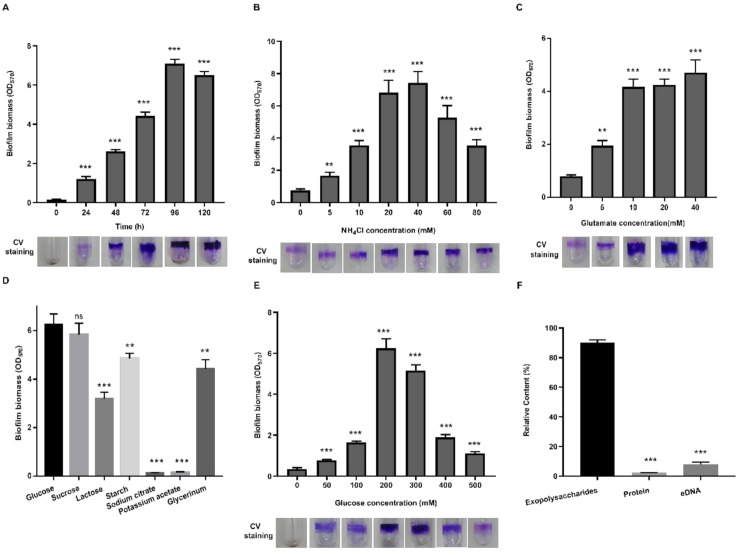
Effects of different nitrogen and carbon sources on biofilm formation of *P. polymyxa.* WLY78. (**A**) Biofilm formation of *P. polymyxa* WLY78 after cultivation in nitrogen-free medium supplemented with 200 mM glucose and 20 mM NH_4_Cl for different times (h). (**B**) Effect of NH_4_Cl concentration on biofilm biomass. (**C**) Effect of glutamate concentrations on biofilm biomass. (**D**) Effect of different carbon sources on biofilm biomass. (**E**) Effect of glucose concentration on biofilm biomass. (**F**) The relative content of each component in extracellular matrix of *P. polymyxa* WLY78 grown in nitrogen-free medium supplemented with 200 mM glucose and 20 mM NH_4_Cl for 60–72 h. “CV” means crystal violet. Results are representative of at least three independent experiments. Error bars indicate SD. Ns indicates nonsignificant. ** indicates *p* < 0.01. *** indicates *p* < 0.001.

**Figure 2 microorganisms-09-00289-f002:**
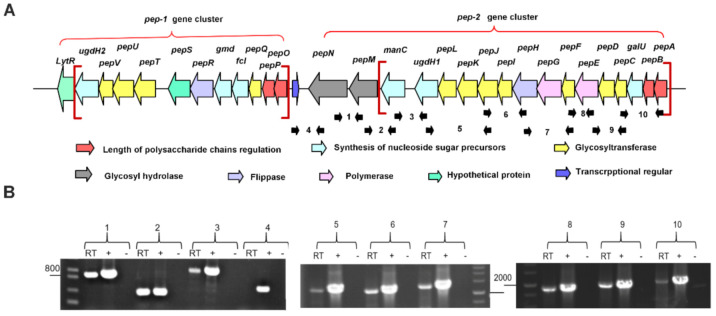
Schematic genetic organization and co-transcription verification of polysaccharide biosynthesis gene clusters. (**A**) Schematic genetic organization of two polysaccharide biosynthesis gene clusters in *P. polymyxa* WLY78. The *pep-1* cluster comprises 12 putative genes and the *pep-2* gene cluster contains 17 putative genes. The interval region between the *pep-1* cluster and the *pep-1* cluster is ~2 kb. The brace indicates the knockout genes. Different colors indicate different functions of genes. Black rows indicate primers used for RT-PCR. (**B**) Confirmation of each of the *pep-1* cluster and the *pep-2* cluster being organized in an operon by RT-PCR. Result of RT-PCR reactions with RNA from *P. polymyxa* WLY78 in agarose gel. The numbers on the top of the gel correspond to the numbers marked schematically in the outline given above. (+), genomic DNA served as a positive control. (−), RNA template was used as the negative control.

**Figure 3 microorganisms-09-00289-f003:**
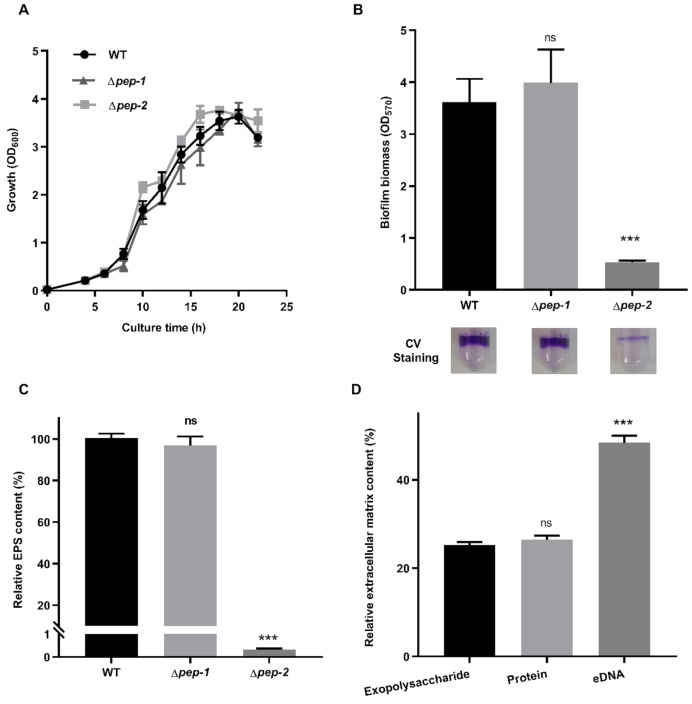
Comparison of wild-type (WT) *P. polymyxa* WLY78, ∆*pep-1* mutant and ∆*pep-2* mutant in terms of growth rate, biofilm biomass and EPS content. (**A**) The growth curves of WT, ∆*pep-1* and ∆*pep-2* in nitrogen-free liquid medium supplemented with 200 mM glucose and 20 mM NH_4_Cl. (**B**) Biofilm formation of WT, ∆*pep-1* and ∆*pep-2* after cultivation for 96 h in the same medium as described above. (**C**) The EPS content of WT, ∆*pep-1* and ∆*pep-2* in extracellular matrix. (**D**) The relative content of each component in the extracellular matrix composition of ∆*pep-2*. Results are representative of at least three independent experiments. Error bars indicate SD. Ns indicates nonsignificant. *** indicates *p* < 0.001.

**Figure 4 microorganisms-09-00289-f004:**
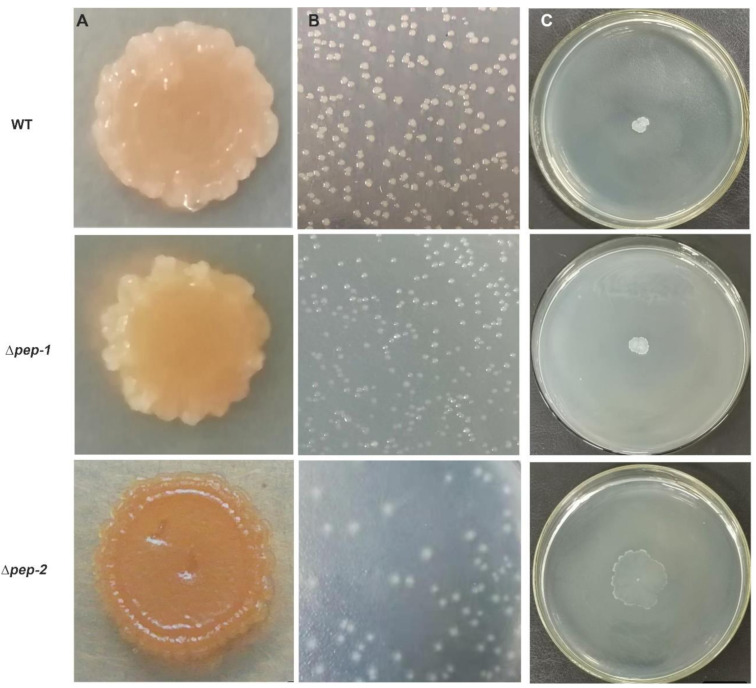
The colony biofilm, colony morphology and motility of *P. polymyxa* WLY78 (WT), *pep-1* and *pep-2* gene cluster. (**A**) Colony biofilm of WT, ∆*pep-1* and ∆*pep-2* after 120 h of cultivation. (**B**) Colony morphology of WT, ∆*pep-1* and ∆*pep-2* after 18 h of cultivation. (**C**) Twitching motility assays of WT, ∆*pep-1* and ∆*pep-2* after 24 h of cultivation at 30 °C. Results are representative of at least three independent experiments.

**Figure 5 microorganisms-09-00289-f005:**
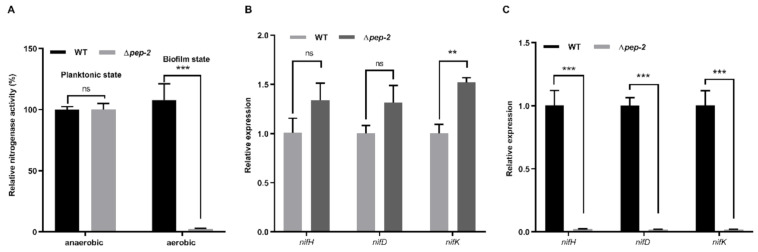
Effect of biofilm formation defect on nitrogenase activity under aerobic conditions. (**A**) The influence of biofilm on the nitrogenase activity under anaerobic condition in the planktonic state or under aerobic conditions in the biofilm state. (**B**) qRT-PCR analysis of the relative mRNA levels of the *nifHDK* genes in the WT and ∆*pep-2* strains under anaerobic condition in the planktonic state. (**C**) qRT-PCR analysis of the relative mRNA levels of the *nifHDK* genes in the WT and ∆*pep-2* strains under aerobic condition in the biofilm state. Relative nitrogenase activity and relative mRNA levels of each sample were normalized to those of the WT sample. Results are representative of at least three independent experiments. Error bars indicate SD. Ns indicates nonsignificant. ** indicates *p* < 0.01. *** indicates *p* < 0.001.

**Figure 6 microorganisms-09-00289-f006:**
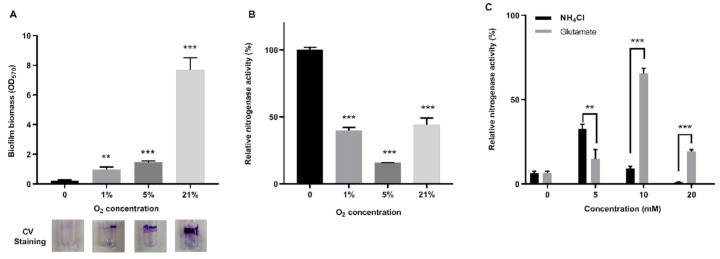
Effects of nitrogen and oxygen on nitrogenase activity in biofilm. (**A**) Effect of different O_2_ concentrations on biofilm formation. (**B**) Effect of O_2_ level on the nitrogenase activity of *P. polymyxa* WLY78 in the biofilm state. (**C**) Effect of different concentrations of NH_4_Cl or glutamate on nitrogenase activity of *P. polymyxa* WLY78 in the biofilm state. Relative nitrogenase activity of each sample was normalized to that of sample under the anaerobic environment in the biofilm state. Results are representative of at least three independent experiments. Error bars indicate SD. ** indicates *p* < 0.01. *** indicates *p* < 0.001.

**Figure 7 microorganisms-09-00289-f007:**
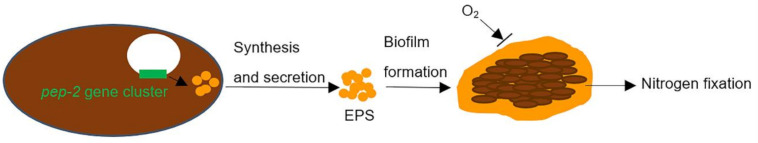
Model of biofilm protection for nitrogenase against O_2_ in *P. polymyxa* WLY78. The *pep-2* gene cluster produces EPS that was secreted to the outside of the cell. Then, the cells of *P. polymyxa* WLY78 are enclosed by EPS to form biofilms. Biofilm provides a microaerobic environment for nitrogenase to fix nitrogen.

**Table 1 microorganisms-09-00289-t001:** The predicted functions of the 29 genes within the *pep-1* and *pep-2* clusters in *P. polymyxa* WLY78.

Predicted Gene Products	Length [aa]	Predicted Function	Predicted Protein Family	Identity with Those of *P. polymyxa* DSM 365
*pep-2* gene cluster				
PepA (GM004746)	247	Capsular biosynthesis protein	YveK	100%
PepB (GM004745)	212	tyrosine protein kinase	P-loop_NTPase	100%
GalU (GM004744)	300	UTP-glucose-1-phosphate uridilytransferase	Glyco_tranf_GTA_type (Glycosyltransferase family A)	100%
PepC (GM004743)	232	undecaprenyl-phosphate galactose phosphotransferase	Bac_transf (Bacterial sugar transferase)	99.57%
PepD (GM004742)	311	undecaprenyl-phosphate galactose phosphotransferase	Glyco_tranf_GTA_type	99.36%
PepE (GM004741)	445	polysaccharide polymerase	Wzy_C	99.78%
PepF (GM004740)	251	UDP-N-acetyl-D-mannosaminuronic acid transferase	Glycosyltransferase WecG/TagA	100%
PepG (GM004739)	480	polysaccharide polymerase	Wzy_C	99.38%
PepH (GM004738)	472	flippase	MATE (Multidrug and toxic compound extrusion) like	99.79%
PepI (GM004737)	276	glycosyltransferase GT2	Beta4 Glucosyltransferase	99.64%
PepJ (GM004736)	395	glycosyltransferase GT4	Glycosyltransferase_GTB-type	100%
PepK (GM004735)	382	glycosyltransferase GT1	Glycosyltransferase_GTB-type	100%
PepL (GM004734)	354	glycosyltransferase GT4	Glycosyltransferase_GTB-type	100%
UgdH1 (GM004733)	445	UDP-glucose 6-dehydrogenase	NADB[NAD(P)H/NAD(P)(+) binding]_Rossmann	100%
ManC (GM004732)	458	mannose-1-phosphate guanylyltransferase	Glyco_tranf_GTA_type	99.56%
PepM (GM004731)	535	glycoside hydrolase	Glycosyl hydrolase family 26	96.45%
PepN (GM004730)	741	glycoside hydrolase	CRISPR/Cas system-associated protein Cas4	97.98%
***pep-1* gene cluster**				
PepO (GM004727)	249	polysaccharide chain length regulation protein	YveK	97.59%
PepP (GM004726)	228	tyrosine protein kinase	P-loop_NTPase	98.68%
PepQ (GM004725)	192	undecaprenyl-phosphate glucose phosphotransferase	Bac_transf	98.96%
Fcl (GM004724)	312	GDP-L-fucose synthase	NADB_Rossmann	98.40%
Gmd (GM004723)	329	GDP-mannose 4,6-dehydratase	NADB_Rossmann	99.39%
PepR (GM004722)	447	putative flippase	MATE_like	99.78%
PepS (GM004721)	420	putative flippase	No domains	100%
PepT (GM004719)	416	glycosyltransferase GT1	Glycosyltransferase_GTB-type	96.88%
PepU (GM004718)	405	glycosyltransferase GT1	Glycosyltransferase_GTB-type	97.78%
PepV (GM004717)	269	UDP-N-acetyl-D-mannosaminuronic acid transferase	Glycosyltransferase WecG/TagA	95.54%
UgdH2 (GM004716)	456	UDP-glucose 6-dehydrogenase	NADB_Rossmann	99.12%
LytR (GM004715)	325	transcriptional regulator	LytR_cpsA_psr	-

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
