# Peer review of "Effects of an EPS Biosynthesis Gene Cluster of Paenibacillus polymyxa WLY78 on Biofilm Formation and Nitrogen Fixation under Aerobic Conditions"

_microorganisms, 2021, doi:10.3390/microorganisms9020289_

Round 1

Reviewer 1 Report

Please see my comments and suggestions in the pdf document. Thanks.

Reviewer 2 Report

Dear authors,

I read and check your manuscript with much interest. Overall, yours is good, but I recommend you to revise/clarify me some points, so that your manuscript will attract valuable readers much more.

  • Please, explain why you formed biofilms in tube instead microtiter plates (Section 2.3).
  • Explain the importance of study the Paenibacillus in terms of biofilm formation and EPS biosynthesis. It was not clear for me. This information should appear clearer in the manuscript.
  • Explain the contribution of the results in terms of their application in different biotechnological areas.
  • In material and methods section, I think you can added information about the statistical analysis performed.
  • Line 58, It seems like you forgot a point after the “spp.”
  • Line, 51, 67, 207 you can change Paenibacillus polymyxa by P. polymyxa.
  • Line 129, you can change NH4Cl by NH4Cl.

Positive points:

  • Figures are very attractive, clean, clear with statistical analysis well represented.
  • The manuscript is well written with essential information.
